# Cortical and Trabecular Bone Stress Assessment during Periodontal Breakdown–A Comparative Finite Element Analysis of Multiple Failure Criteria

**DOI:** 10.3390/medicina59081462

**Published:** 2023-08-15

**Authors:** Radu Andrei Moga, Cristian Doru Olteanu, Stefan Marius Buru, Mircea Daniel Botez, Ada Gabriela Delean

**Affiliations:** 1Department of Cariology, Endodontics and Oral Pathology, School of Dental Medicine, University of Medicine and Pharmacy Iuliu Hatieganu, Str. Motilor 33, 400001 Cluj-Napoca, Romania; ada.delean@umfcluj.ro; 2Department of Orthodontics, School of Dental Medicine, University of Medicine and Pharmacy Iuliu Hatieganu, Str. Avram Iancu 31, 400083 Cluj-Napoca, Romania; 3Department of Structural Mechanics, School of Civil Engineering, Technical University of Cluj-Napoca, Str. Memorandumului 28, 400114 Cluj-Napoca, Romania; marius.buru@mecon.utcluj.ro (S.M.B.); mircea.botez@mecon.utcluj.ro (M.D.B.)

**Keywords:** bone, bone loss, orthodontic force, finite element analysis, orthodontic movement

## Abstract

*Background and Objectives:* This numerical analysis investigated the biomechanical behavior of the mandibular bone as a structure subjected to 0.5 N of orthodontic force during periodontal breakdown. Additionally, the suitability of the five most used failure criteria (Von Mises (VM), Tresca (T), maximum principal (S1), minimum principal (S3), and hydrostatic pressure (HP)) for the study of bone was assessed, and a single criterion was identified for the study of teeth and the surrounding periodontium (by performing correlations with other FEA studies). *Materials and Methods:* The finite element analysis (FEA) employed 405 simulations over eighty-one mandibular models with variable levels of bone loss (0–8 mm) and five orthodontic movements (intrusion, extrusion, tipping, rotation, and translation). For the numerical analysis of bone, the ductile failure criteria are suitable (T and VM are adequate for the study of bone), with Tresca being more suited. S1, S3, and HP criteria, due to their distinctive design dedicated to brittle materials and liquids/gas, only occasionally correctly described the bone stress distribution. *Results:* Only T and VM displayed a coherent and correlated gradual stress increase pattern for all five movements and levels of the periodontal breakdown. The quantitative values provided by T and VM were the highest (for each movement and level of bone loss) among all five criteria. The MHP (maximum physiological hydrostatic pressure) was exceeded in all simulations since the mandibular bone is anatomically less vascularized, and the ischemic risks are reduced. Only T and VM displayed a correlated (both qualitative and quantitative) stress increase for all five movements. Both T and VM displayed rotation and translation, closely followed by tipping, as stressful movements, while intrusion and extrusion were less stressful for the mandibular bone. *Conclusions:* Based on correlations with earlier numerical studies on the same models and boundary conditions, T seems better suited as a single unitary failure criterion for the study of teeth and the surrounding periodontium.

## 1. Introduction

The bone structure and periodontal ligament (PDL) are the supporting tissues of the tooth and are subjected to various amounts and forms of stresses during orthodontic treatment [1,2]. Two types of bone can be distinguished: the cortical and the trabecular/cancellous bone, which, biomechanically, should be analyzed as a continuum [1,2,3,4,5,6,7,8]. The trabecular component holds the bone marrow and vascular vessels and has a higher regeneration potential than cortical bone [9]. Both components are anatomically anisotropic materials, with trabecular bone being a highly porous mineralized material while cortical bone is a highly mineralized compact structure [1]. Cortical bone has the function of structural support for surrounding dental tissues and the protection of trabecular/cancellous bone [5].

Bone as a continuum and as a single-stand structure possesses a high adaptation ability to alter its geometry to provide the strongest structure possible with a minimum amount of tissue [7]. The bone structure can also absorb and dissipate energy/stresses and elastically deform, preventing fracture and/or destruction [7]. Its internal structural micro-architecture allows for microcracks/damage (i.e., linear, and diffuse microcracks, and microfractures) and time to heal [7]. Linear microcracks appear as a response to compressive stresses (older age, more brittleness), the diffuse microdamage as a response to tensile stresses (younger age, more ductileness), while microfractures as a response to shear stresses (older age, mixed ductile brittleness, mostly in trabecular bone) [7]. Thus, from a biomechanical engineering perspective, the bone structural behavior depends on the applied loads acting as a ductile material with a certain brittle flow mode [2,7,10,11,12]. 

The orthodontic stresses from the tooth are transmitted through the PDL to the bone, while the display areas are also influenced by the tissular anatomy and integrity [8]. Periodontal breakdown is found in orthodontic patients, affecting the biomechanical behavior of the tooth and surrounding support tissues, with higher tissular amounts of stress appearing along with bone loss [2,10,11,12]. In the bone structure, Burr et al. [7] reported microcracks and microdamage near the resorption and remodeling areas and a decrease in fracture risks in their presence (due to internal micro-architectural changes), which influence the structural stress display. It must be emphasized that the biomechanical behavior in both the intact and reduced periodontium is multifactorial, which depends on the cortical and trabecular structural continuum, material, and structural properties [6].

No studies investigating the mandibular bone stress distribution in a gradual periodontal breakdown under orthodontic loads were found, despite multiple bone-implant FEA (finite element analysis) research studies with a focus on the implant and the surrounding bone [1,5,8,13,14,15,16,17,18].

Only three reports were found to assess the stress distribution both in the tooth and its intact surrounding support system (0.35–0.5 N of tipping [19,20]; 10 N of intrusion, 3 N of tipping and translation [21]); these showed various qualitative and quantitative results but no correlation with the maximum physiological hydrostatic pressure (MHP) and/or failure criteria type of the analyzed material. These reports employed both ductile (Von Mises) and brittle (maximum principal stress) failure criteria and supplied results that did not entirely match the clinical data.

The orthodontic biomechanical behavior of bone is influenced by the anatomy of tissues, materials, magnitude, and the quantity and quality of the bone [1,14]. There are several tools to analyze the biomechanical behavior of bone and teeth, including in vitro assays (photoelastic stress analysis, static/dynamic mechanical fracture tests) and numerical simulations (finite elements analysis) [1]. FEA is the only method that allows the individual analysis of each component of a structure, providing accurate results if the input data (anatomical accuracy and loading conditions) are correct [1,13]. Only a numerical simulation such as FEA allows for correct biomechanical studies that assess and predict stress distribution in living dental tissues [13,14,17,22].

FEA accuracy also depends on the selection of proper and adequate failure criteria. There are multiple failure criteria, each specially designed to better describe the biomechanical behavior of a certain type of material: brittle-maximum S1 tensile and minimum S3 compressive principal stresses, ductile-Von Mises (VM) overall/equivalent and Tresca (T) shear stress, and liquid/gas-hydrostatic pressure (HP) [2,10,11,12]. The main difference between these types of materials is related to the way they deform under loads (yielding materials theory) [2,10,11,12]. The ductile materials suffer from various forms of recoverable deformations, returning to their original form after the force effect has ceased [2,10,11,12]. The brittle materials, when subjected to various loads, suffer from various degrees of plastic non-recoverable deformations, with modification of their original shape and dimensions (necking and buckling effects) before their fracture and destruction [2,10,11,12]. Hydrostatic pressure describes a specific physical state where there is no shear stress (which is not adequate for solid materials as ductile and brittle) [2,10,11,12]. This approach, based on the assessment of material type when performing the FEA analysis, has not been found in other studies (except in our team’s earlier studies [2,10,11,12]) despite its importance for the accuracy of results [22]. Moreover, there are no FEA studies of bone to compare various failure criteria and to select the most adequate one based on the results.

The dental tissues (dentine, cement, dental pulp, neurovascular bundle, PDL, bone, and stainless-steel bracket) are all considered to resemble ductile materials with a certain brittle flow mode [2,10,11,12]. Only enamel is a brittle material due to its internal micro-architecture [23,24]. Nevertheless, since it represents only an extremely small percentage of the entire volume of dental tissues, and the entire structure behaves as ductile, the adequate and acceptable failure criteria is that of ductile materials [2,10,11,12,17].

Bone, when subjected to internal stress, undergoes a certain amount of recoverable elastic deformations because of the PDL stress transmitted to the bone, beyond which microfractures appear and bone loss results [14]. According to the engineering composite beam theory, when materials with different elastic modulus (cortical and trabecular bone, PDL and dentine; Table 1) interact and are subjected to a load, the highest stress is located at the first point of contact (i.e., bone cervical third) [14]. Hooke’s law states that the deformation of materials depends on their elastic modulus; the higher the modulus, the smaller the deformation [14,22]. In the tooth and surrounding support system, the periodontal ligament, followed by bone, suffers the highest deformation [10,11,12,14].

Most bone-implant FEA studies employed the adequate VM failure criteria in intact bone, reporting stress concentrations in the cortical component located in cervical third areas, while in the trabecular component, these occurred in a broader area [1,5,8,13,14,15,16,17,21]; however, they did not address the suitability issues (VM is more suited for homogeneous materials, while bone is non-homogenous). There are biomechanical reports suggesting that the shear stress produced by occlusal loadings contributes to bone resorption around the implants [13]. However, there were no studies found assessing the periodontal breakdown influence over the stress distribution in bone.

For avoiding ischemia, necrosis, and further periodontal loss, the physiological maximum hydrostatic pressure of 16 KPa [2,10,11,12] should not be exceeded, especially in the well-vascularized dental tissues and dental tissues that are easily deformable under stress (i.e., PDL, dental pulp, and the neuro-vascular bundle (NVB)). However, in the less deformable and vascularized tissues (i.e., dentine, bone), amounts of stress higher than the MHP could appear without significant tissue losses. However, these amounts of stress should not exceed the maximum tensile, shear, and compressive strength of each material (which never occurs in clinical daily practice). 

Nevertheless, in orthodontic biomechanics, the PDL is the triggering factor for the orthodontic movements (due to circulatory disturbances) inducing bone remodeling. If these circulatory disturbances are severe and last for a longer period, the inevitable ischemia will lead to necrosis and tissue loss. Usually, in intact periodontium, the orthodontic forces from daily practice are light [25] and up to 1.5 N (approx. 150 gf) [2,10,11,12]. Nevertheless, there is little information about the orthodontic forces that can be safely applied in the reduced periodontium [10,11,12]. Earlier studies from our group reported for the PDL, dental pulp, and neurovascular bundle a reduction of applied forces for an 8 mm reduced periodontium to avoid exceeding the MHP. The areas of higher stress were reported to be the cervical third of PDL, with less stress in the apical third, where the NVB is found. Nevertheless, the issues of MHP and correlations with the highly vascularized dental tissues (as PDL, pulp, and NVB) should be approached in a bone study.

In the dental field, the FEA numerical method is well represented in many studies of PDL and implants. The mostly used failure criteria are the Von Mises (VM) overall/equivalent stress [18,19,20,21,22,26,27,28], Tresca (T) maximum shear stress [2,10,11,12], maximum principal S1 tensile stress [19,22,29,30,31,32], minimum principal S3 compressive stress [22,29,30,32,33,34,35], and hydrostatic pressure (HP) [36,37,38,39,40]. However, a recurrent issue in these studies (except in ours [2,10,11,12]) is the lack of correlations between the inner anatomical micro-structure, the material type resemblance, the criteria suitability, the coherent biomechanical behavior resembling to clinical knowledge, the quantitative results correlated with the physiological maximum hydrostatic pressure (MHP), the orthodontic force dissipation and absorption ability, and the biomechanically correct stress display. Thus, FEA is still approached with care since results are often supplied that contradict clinical knowledge [2,10,11,12].

In the engineering field, the FEA simulations are extremely accurate since all the above issues related to diverse types of correlations are addressed and the adequate failure criteria and correct input data have been defined. To have the same accuracy of the FEA method in dental studies, it is necessary that a single failure criterion addressing all above correlations is assessed to be scientifically accurate and providing results correlated with clinical and theoretical knowledge [2,10,11,12]. Previous studies from our group reported the ductile resemblance and showed that only VM and T criteria met all the above expectations, with Tresca proven to be more accurate for the tooth structural components, PDL, and dental pulp with NVB [2,10,11,12]. Thus, the bone FEA study herein completes the data necessary for assessing the general failure criteria for the tooth and surrounding support periodontium. 

The objectives of this FEA analysis are (a) to assess the biomechanical behavior of mandibular bone subjected to light orthodontic forces during a horizontal periodontal breakdown; (b) to assess its suitability for the study in bone of five of the most used failure criteria employed in dental tissue research; (c) to correlate the results with other FEA-related reports of dental tissues to identify a suitable single unitary failure criteria for the study of teeth and the surrounding periodontium.

## 2. Materials and Methods

The numerical analysis herein is part of a larger stepwise research project (clinical protocol 158/02.04.2018) continuing the investigation of biomechanical behavior of teeth and surrounding periodontal structure during orthodontic movements and various levels of periodontal breakdown.

The earlier analyses of this project, with a focus on the dental pulp, neuro-vascular bundle (NVB), periodontal ligament (PDL), dentine and enamel, were conducted using the same models, boundary conditions, and physical properties as herein [2,10,11,12].

The 405 FEA numerical simulations were conducted over eighty-one 3D mandibular models holding the second lower premolar obtained from nine patients (4 males/5 females, mean age 29.81 ± 1.45). The selected convenience sample size of nine was acceptable for the accuracy of the results since most of the earlier FEA studies employed a sample size of one (one patient, one 3D model, and few simulations) [1,2,5,8,10,11,12,13,14,15,16,17,18,19,20,21,22,26,27,28,29,30,31,32,33,34,35,36,37,38,39,40]. The research project inclusion criteria were the presence of non-inflamed periodontium and various levels of bone loss, an intact arch and second premolar tooth structure, lack of endodontic treatment and malposition in the region of interest, indication of orthodontic treatment, and regular follow-up. All the situations that were not covered by the above criteria were considered to be exclusion criteria (especially the lack of arch integrity, tooth malposition, more than 8 mm bone-loss cases, and inflamed periodontium). 

The lower mandibular region containing the premolars and first molar was examined by CBCT (ProMax 3DS, Planmeca, Helsinki, Finland), obtaining images of various shades of gray, with a voxel size of 0.075 mm.

The radiological Hounsfield gray shade units present on the DICOM slices were examined to identify the dental tissues. The anatomically accurate reconstruction of the tissues was performed through manual segmentation since the automated software algorithm did not accurately identify all the structures. Thus, the enamel, dentine, dental pulp, neurovascular bundle, periodontal ligament, cortical and trabecular bone were reconstructed in 3D (Figure 1). The reconstruction software was Amira 5.4.0 (Visage Imaging Inc., Andover, MA, USA). The base of the bracket, assumed to be of stainless steel, was reconstructed on the vestibular side of the enamel crown. Since the separation of the dentine and the cementum was impossible, and due to similar physical properties, the entire dentine–cementum structure was reconstructed as dentine (Table 1). The PDL had a variable thickness of 0.15–0.225 mm and included the NVB of the dental pulp in the apical third. The 3D models guarded only the second lower premolar, while the other tooth structures were replaced by cortical and trabecular bone. The missing bone and PDL (which were found in the cervical third) were reconstructed as closely as possible to the anatomical reality. Thus, nine models with intact periodontium (one from each patient) were obtained. In each of these models, a gradual horizontal breakdown process (0–8 mm of loss) was simulated by reducing both bone and PDL by 1 mm, obtaining a total of eighty-one models with various levels of bone loss. The 3D intact periodontium models had 5.06–6.05 million C3D4 tetrahedral elements, 0.97–1.07 million nodes, and a global element size of 0.08–0.116 mm. 

The surface of the models, due to the manual segmentation technique, displayed a limited number of element warnings and no element errors (Figure 1). Thus, for one of the models shown in Figure 1K,L, the cortical bone mesh displayed 131 element warnings for 3,417,625 elements (i.e., 0.00383%), while for the trabecular mesh, there were only 70 element warnings for 1,699,730 elements (i.e., 0.00411%). The element warnings and surface anomalies were displayed in non-essential areas since the stress areas were quasi-continuous, and both numerical analysis software allowed the passing of the internal checking algorithms.

The numerical analysis was performed using the Abaqus 6.13-1 (Dassault Systèmes Simulia Corp., Maastricht, The Netherlands). Five of the most-used failure criteria were employed: Von Mises maximum overall/equivalent stress (VM), Tresca maximum shear stress (T), maximum S1 tensile principal stress, minimum S3 compressive principal stress, and hydrostatic pressure (HP). Five orthodontic movements were simulated: intrusion, extrusion, rotation, translation, and tipping, under an applied load of 0.5 N (approx. 50 g) on the base of the bracket (Figure 1). This amount of load was selected since it is a light force that is relatively safe to be applied to both the intact and the reduced periodontium; this also enables the results herein to be correlated with earlier studies with a focus on other dental tissues [2,10,11,12]. 

The assumed boundary conditions were the homogeneity, linear elasticity, and isotropy, like most of the available numerical analyses [1,2,5,8,10,11,12,13,14,15,16,17,18,19,20,21,22,26,27,28,29,30,31,32,33,34,35,36,37,38,39,40]. 

The results of the numerical simulations are presented as color-coded projections of the stress display areas (qualitative, Figure 2, Figure 3, Figure 4, Figure 5 and Figure 6), and quantitative (amounts of stresses, Table 2). The biomechanical behavior displayed by each orthodontic movement and described by each failure criteria were compared and correlated with earlier analysis [2,10,11,12] to determine if one of the failure criteria is better suited for the study of dental tissues. 

## 3. Results

Our analysis involved eighty-one mandibular models in 405 FEA numerical simulations (Figure 2, Figure 3, Figure 4, Figure 5 and Figure 6 and Table 2). No influences due to age, sex, or periodontal status were seen.

From the qualitative point of view (i.e., the color-coded stress display from Figure 2, Figure 3, Figure 4, Figure 5 and Figure 6), both the Tresca and Von Mises failure criteria displayed similar color-coded projections for all five orthodontic movements in both the intact and the reduced periodontium. S1, S3, and HP criteria displayed various color-coded projections of the stress distribution and no visible constant pattern: sometimes resembling T and VM, sometimes showing unusual biomechanical behavior (i.e., different from acknowledged clinical reality). Moreover, the boundary conditions (i.e., encastered base of the model, with zero displacements) significantly influenced the biomechanical behavior of the models when S1, S3, and HP criteria were employed.

From the quantitative point of view (Table 2), the unusual biomechanical behavior was seen in S1 and S3 criteria in the rotation, translation, and tipping movements during the periodontal breakdown with a variable decrease/increase of 0–4 mm and 4–8 mm bone loss. The HP criteria displayed a decrease in the amounts of stress for intrusion, extrusion and translation, and an increase for rotation and tipping. No stress pattern and/or correlations were observed among the S1, S3, and HP criteria despite investigating the same models under the same conditions as T and VM. Only T and VM criteria displayed a constant expected increase pattern during the periodontal breakdown process for all five movements, confirming the acknowledged clinical biomechanical behavior. The rotation movement seemed to be the most bone-stressful of all five criteria. Nevertheless, T and VM displayed rotation and translation, closely followed by tipping, as stressful movements, while intrusion and extrusion were less stressful for the mandibular bone. 

All quantitative stresses displayed in the mandibular bone for all five failure criteria exceeded the MHP of 16 KPa in both the intact and the 8 mm reduced periodontium. The highest amount of stress was produced by the rotation at 8 mm of bone loss (335 KPa [0.335 MPa]) under the T criteria. The vestibular cervical third stress was the highest for all criteria, movements, and levels of bone loss. All stresses were lower than the acknowledged physical properties of bone: cortical bone compressive modulus, 16.7 GPa, and compressive strength, 157 MPa; trabecular/cancellous bone compressive modulus, 0.155 GPa, and compressive strength, 6 MPa [1,5,8,13,14,15,16,17,18].

### 3.1. Extrusion (Figure 2)

VM (overall stress) and T (shear stress) criteria displayed similar color-coded projections in both the intact and the reduced periodontium. The stressed areas were found in the cervical third of the alveolar wall where the bone is thinner. The cervical third stress location was maintained during the entire periodontal breakdown, with a proportional increase correlated with the bone loss. On the lingual side of the mandibular bone model, due to boundary conditions (base of the model had zero displacements), reduced amounts of stress were displayed. The S1 maximum principal tensile stress displayed from 4 to 8 of bone loss, which was an unusual stress increase both on the vestibular and the lingual surfaces (for such a small extrusion force of 0.5 N). The S3 minimum principal compressive stress displayed an unusual maximum moment of stress increase at 4 mm of loss (on both the vestibular and lingual sides, quantitatively visible in Table 2), despite the cervical third stress having a similar display as T and VM. The HP criteria displayed the same unusual biomechanical stress increase at 4 mm of bone loss, with a further apical third increase on both the vestibular and lingual sides at 8 mm of loss (due to boundary conditions). All quantitative values were higher than MHP.

### 3.2. Intrusion (Figure 3)

T and VM criteria displayed higher cervical third stress on the vestibular side of the model during the entire periodontal breakdown simulation, with a gradual increase in the apical third from 4–8 mm of loss. The lingual side stress at the base of the model appears to correlate with the progression of bone loss. The quantitative values are like those seen in the extrusion movement. S1 and S3 criteria showed a visible unusual stress display, with higher stresses on both the vestibular and lingual sides, and a decrease in stress from 0–4 mm followed by an increase from 4–8 mm of loss. HP criteria displayed a quantitative cervical stress decrease correlated with bone loss and a light increase in the apical third and base of the model.

### 3.3. Rotation (Figure 4)

T and VM displayed a cervical third stress increase correlated with the periodontal breakdown on the vestibular side of the bone. Both criteria display stresses found at the base of the models due to the assumption of zero displacements. Rotation produced the highest amounts of stress among all five movements. Both S1 (tensile) and S3 (compressive) criteria displayed unusual extended stress areas for only 0.5 N of rotation and a decrease in quantitative vestibular cervical third stress from 0–4 mm followed by an increase up to 8 mm of loss. HP stress displays mostly vestibular cervical third stress areas (and stress at the base of the model lingual side), and with lower amounts of stress than T and VM.

### 3.4. Tipping (Figure 5)

Vestibular side cervical third stress was shown during the periodontal breakdown by both T and VM criteria. On the lingual and vestibular sides close to the base of the model, visible stresses were displayed due to assumed boundary conditions. Both criteria displayed a quantitative increase in stresses correlated with the bone loss levels. S1 criteria displayed an unusual stress pattern with less stress in the cervical third and more on the vestibular side close to the base of the model. S3 displayed the highest compressive stress in the cervical third for intact periodontium, while visible stresses (despite the reduced applied force of 0.5 N) were also seen on the lingual side close to the base of the model. Quantitatively, S3 displayed a stress increase correlated with bone loss. S1 criteria displayed an unusual decrease in stress in the cervical third from a 0–4 loss followed by an increase from a 4–8 mm loss. HP criteria displayed a stress increase in both the vestibular cervical third and the lingual side close to the model base.

### 3.5. Translation (Figure 6)

Both T and VM criteria displayed higher stresses in the vestibular cervical third, with a reduction of the extension correlated with the progress of the periodontal breakdown. Both criteria displayed a correlated quantitative stress increase with the bone loss levels. Both S1 and S3 criteria showed an unusual stress display, and a quantitative decrease from 0–4 mm followed by an increase from 4–8 mm of loss. HP criteria showed almost no stress areas and a constant quantitative stress decrease.

For all five movements and periodontal breakdown levels, the S1, S3 (specific for brittle materials), and HP (specific for liquids/gas) criteria displayed unusually extended and variable stress areas for such a small force (0.5 N/ approx. 50 gf). The quantitative variable stress increase/decrease at the 0–4 and 4–8 mm bone levels and the lack of any biomechanical pattern or correlations seemed to confirm the reduced accuracy of these three criteria. The base of the models was assumed to have no displacements (simulating the stiffness of the mandibular bone); thus, areas of stress found close to the base of the models were expected (similar to other numerical analysis). However, S1, S3 and HP displayed an unusual extension of these stress areas compared with T and VM criteria.

T and VM for all five movements and bone levels displayed a coherent and correlated gradual stress increase pattern (Table 2), the expected biomechanical behavior (Figure 2, Figure 3, Figure 4, Figure 5 and Figure 6), and a limited stress area at the base of the model. The MHP was exceeded in all simulations (e.g., 6.7–16 times in the vestibular cervical third for T and VM criteria), as expected, since the mandibular bone is anatomically less vascularized, and the ischemic risks are reduced. 

The quantitative values provided by T and VM were the highest among all the five criteria. The differences between the T and VM values were approx. 15%, thus falling within the range reported in the literature of 15–30%. 

When the qualitative and quantitative results are correlated, T and VM seemed to be more adequate for the study of bone biomechanical behavior than the other three criteria, with T more suited due to its mathematical design for non-homogenous ductile structures with a certain brittle flow mode (dental tissues included [2,10,11,12]).

## 4. Discussion

The present study assessed the biomechanical behavior of bone as a single-stand/continuum structure (with both cortical and trabecular/cancellous components) when subjected to stresses projected by 0.5 N of orthodontic force during periodontal breakdown. The experiments were performed through 405 FEA simulations on over eighty-one 3D mandibular models with the second lower premolar included, employing five of the most used failure criteria, with the aim of finding the most suitable criteria for bone analysis. It must be emphasized that this is the only study investigating these issues. Moreover, by correlating the results herein with those from earlier studies [2,10,11,12], we aimed to identify a single general failure criterion that can be used for the numerical analysis of dental tissues.

During the periodontal breakdown simulations under the five orthodontic movements, only the T (Tresca) and VM (Von Mises) criteria displayed a coherent and correlated pattern, both qualitatively and quantitatively, and a biomechanical behavior resembling that in vivo (Figure 2, Figure 3, Figure 4, Figure 5 and Figure 6 and Table 2). The other three criteria (S1—maximum tensile, S3—minimum compressive, and HP—hydrostatic pressure) displayed no visible biomechanical behavioral pattern despite investigating the same models and boundary conditions (Figure 1 and Table 1) as T and VM. Between T and VM, due to their mathematical design for different types of internal micro-structures, T seemed to be more suitable than VM (i.e., similar qualitative color-coded projections but with 15% increase in quantitative values, in agreement with other reports [2,10,11,12]). Both VM and T were designed for ductile materials, the major difference being that T better describes the behavior of non-homogenous materials, while VM is more adequate for homogenous ones [2,10,11,12]. It must be emphasized that bone has a biomechanical behavior resembling ductile materials (with a certain brittle flow mode), displaying various recoverable elastic deformations when subjected to stresses that totally return to their original form after the forces disappear [1].

The S1 (tensile) and S3 (compressive) failure criteria are adequate for brittle materials (little or no deformations before fracture/destruction), and from the biomechanical point of view, these should display a complementary behavioral pattern and correlation since the tension and compressions are two physical transitional phases of deformation [22]. Nevertheless, no such correlation was seen, suggesting that both S1 and S3 do not meet the necessary accuracy for the study of mandibular bone.

The HP was specially designed for liquids and gas where there are no shear stresses during their behavior [2,10,11,12]. With HP being suitable for liquids (e.g., circulatory fluids), and since the mandibular bone has a reduced vascularization (being only a percentage of the entire tissue volume), its failure criteria are even less adequate than those of S1 and S3, visible in Figure 2, Figure 3, Figure 4, Figure 5 and Figure 6.

The FEA analysis of dental tissues needs a single unitary failure criterion that better describes the stress display and provides quantitative results that correlate with clinical data [1]. Both T and VM seem to meet these criteria. In earlier studies [2,10,11,12] of this issue, both criteria were reported to be suitable for the periodontal ligament, dental pulp, and neuro-vascular bundle, and tooth (dentine/cementum, enamel, and stainless-steel bracket), with T being more adequate than VM.

A failure criterion must also provide results that correlate with the maximum hydrostatic pressure values (about 80% of the systolic pressure) found in the dental tissues, which, if exceeded, would lead to ischemia, necrosis, further periodontal loss, and internal and external orthodontic resorptive processes. Both T and VM criteria were reported to supply quantitative results that met this request, while the other three criteria failed [2,10,11,12].

There are multiple FEA bone-implant studies on stress distribution [1,5,8,13,14,15,16,17,18] using uniaxial loading, the VM criteria, and that report concentrations of stress in the cervical third of the bone around the implant, in line with our findings. However, due to the biomechanical behavioral, differences caused by the lack of a PDL and much higher amounts of applied loads than herein (3–10 N [5,18]; 40–800 N [1,8,13,14,15,16,17] vs. 0.5 N), the quantitative results cannot be compared despite the similar boundary conditions and failure criteria. Nonetheless, the qualitative results (the color-coded projection of the VM overall stress) could be correlated with those herein, displaying similar results for the stress distribution areas.

Only three FEA tooth–bone studies of stress distribution were found [19,20,21], with the same boundary conditions and failure criteria as herein. Merdji et al. [21] (single intact periodontium models, 10 N of intrusion, 3 N of tipping and translation, lower third molar, intact bone, VM criteria, 0.25–1 mm global element size, 142305 elements of the mandibular bone) reported a similar cervical third stress display for all three movements as herein. Nonetheless, some differences are also visible since the Merdji et al. [21] model was of the third molar with a different anatomical geometry (equal thickness of the lingual and vestibular bone, three roots) vs. the second premolar herein (vestibular wall of the alveolar process much thinner than the lingual one especially during periodontal breakdown, two roots). The quantitative amounts of cervical stress were 10.5 MPa for 10 N of intrusion, 11.5 MPa for 3 N of tipping, and 16.83 MPa for 3 N of translation [21] vs. 108.34 KPa (0.108MPa) for 0.5 N of intrusion, 106.18 KPa (0.106 MPa) for 0.5 N of tipping, and 135.75(0.135 MPa) for 0.5 N of translation reported herein. These quantitative differences could be due to the differences between the elements’ size and the number of tetrahedral elements of the bone structure (0.25–1 mm/142,305 elements [21] vs. 0.08–0.116 mm/5117355 elements in our analysis).

Using two mandibular models, Field et al. [20] simulated 0.35 N (0.5 N resulting load) of tipping movements (i.e., a single intact periodontium model of 32,812 elements, with incisor, canine, and first premolar, and a single intact periodontium model with canine of 23,565 elements, global element size 1.2 mm, VM, and S1, S3, and HP criteria). The qualitative results (color-coded stress distribution) of Field et al. [20] resembled those herein but with a different color intensity (red high-stress [20] vs. blue-green lower stress in the models herein, which is closer to the clinical biomechanical behavior of a light force). The quantitative results for the bone cervical third stress were 236.3–287.8 KPa VM, 110.1–135.5 KPa S1, and 9.24–(−11.4) KPa S3 [20] vs. 106.18 KPa VM, 53.18 KPa S1, and −171 KPa S3 herein. The differences between Fields et al. [20] and our results come from the modeling issues and the applied forces (e.g., 23565–32812 number of elements and global element size of 1.2 mm vs. 5.06–6.05 million elements, 0.97–1.07 million nodes, and a global element size of 0.08–0.116 mm herein). In additional support of this assumption, Field et al. [20] reported an HP stress of 32 KPa in the apical third of the PDL and a VM stress of 235.5–324.5 KPa in the PDL, amounts that both exceed the 16 KPa of physiological MHP signaling related to increased risks of ischemia, necrosis, and periodontal loss, which contradicts clinical knowledge [25]. These reports of stress [20] are unusually high (for a light orthodontic force of 0.5 N), signaling potential tissue damage which in reality does not occur [25]; thus, these are in need of an explanation. Nonetheless, no correlation with MHP and clinical biomechanical behavior was found in the Field et al. [20] study.

Field et al. [20] also reported a difference between the single tooth and multiple teeth numerical analyses, with higher amounts of stress and larger extension of the stress areas in the multi-tooth system. However, our herein results are lower than those reported by Field et al. [20], and we expect that that same pattern would be followed by a model with multiple teeth.

Similar correlations have been discussed and shown in earlier studies [10,11,12] using same models and boundary conditions as herein, but with a focus on the PDL, dental pulp, NVB, and dentine tissues. By using the same failure criteria comparison as herein, both VM and T criteria were proven to be adequate for dental tissues, with T being more accurate. Moreover, 0.5 N of force was proven to be safely applied to PDL, dental pulp, and NVB in both intact and 8 mm periodontal breakdown, with quantitative results lower than the MHP and correlated with the acknowledged clinical biomechanical behavior [10,11]. For the less vascularized tissues, such as dentine, the quantitative amounts of stress were higher than the MHP, as they biomechanically should be (correlating with the amounts of stress in bone herein), with qualitative stress display areas correlated with those present in vascularized ones (PDL, dental pulp, and NVB) [12]. The results herein, both quantitative and qualitative, are in line with the above approach. 

Shaw et al. [19] (upper incisor, intact periodontium model of 11,924 elements and 20,852 nodes, VM and S1 criteria) reported, for the same five movements (intrusion, extrusion, tipping, translation, and rotations), lower amounts of cervical stress (1.664 KPa for intrusion/extrusion, 0.6 KPa for translation, 0.54 KPa for tipping, and 0.015 KPa for rotation) and comparable stress display areas. We assume that the differences are due to the anatomical accuracy and geometry of the models (incisive vs. premolar, and 507 times fewer elements than herein).

FEA analysis can supply accurate results as in the engineering field if the same requirements are fulfilled (i.e., proper failure criteria, boundary conditions, and physical properties) [2,10,11,12]. The selection of the failure criteria is mandatory for the correctness of both qualitative and quantitative results [22]. A single unitary criterion for the FEA analysis of teeth and the surrounding periodontium is needed, and based on the available data, this could be easily identified [2,10,11,12]. This criterion is scientifically based on the type of internal architecture of the analyzed material (suiting its internal micro-architecture [23,24]) that better describes its biomechanical behavior and provides results that are correlated with the clinical practical data [25] and other numerical analyses [2,10,11,12]. Bone-implant studies [1,5,8,13,14,15,16,17,18] exclusively employed, as adequate, the ductile material resemblance VM failure criteria, while bone–tooth studies [19,20,21] employed multiple criteria: ductile (VM), liquid (HP), and brittle (S1, S3). The PDL studies also employed Von Mises (VM) overall/equivalent stress [18,19,20,21,22,26,27,28], Tresca (T) maximum shear stress [2,10,11,12], maximum principal S1 tensile stress [19,22,29,30,31,32], minimum principal S3 compressive stress [22,29,30,32,33,34,35], and hydrostatic pressure (HP) [36,37,38,39,40]. In all FEA simulations, only the ductile resemblance criteria (VM and T) matched the clinical data [2,10,11,12].

The dental tissues, despite resembling ductile material, have a certain variable amount of brittleness; thus, when brittle material failure criteria such as S1 and S3 are applied, these can sometimes provide both qualitative and quantitative results matching the adequate ductile criteria as in the above studies [19,22,29,30,31,32,33,34,35] and also shown herein in Figure 2, Figure 3, Figure 4, Figure 5 and Figure 6. However, when conducting a larger survey of multiple movements and various bone loss levels, the non-suitability of S1 and S3 becomes more and more visible (as shown by the herein comparisons in Figure 2, Figure 3, Figure 4, Figure 5 and Figure 6 and in our earlier studies [2,10,11,12]). Only one study was found that provided knowledge regarding the differences between the application of various failure criteria and that had similar reports [22] as herein, except for the brittleness of endodontic filing materials. Moreover, this study [22] emphasized the differences that are due to the employment of different brittle failure criteria when analyzing a brittle material and thus the importance of selecting better suited criteria. 

The HP criteria (which was also largely employed in the study of PDL based on the rich proportion of fluids contained in the ligament [36,37,38,39,40]) do not describe the shear state since liquids do not go through this physical state (common engineering knowledge). Moreover, when multiple HP studies are compared, the reports are variable and do not match the clinical practical knowledge.

Studies by Wu et al. [38,39,40] reported various (0.28–3.31 N) optimal forces for the intact PDL subjected to orthodontic movements of the canine, premolar, and lateral incisive. However, despite using the same models and boundary conditions, all three studies reported significant differences for the same movement and tooth (e.g., canine: rotation 1.7–2.1 N [40] and 3.31 N [38]; extrusion 0.38–0.4 N [40] and 2.3–2.6 N [39]; premolar: rotation 2.8–2.9 N [38]), which were all much higher compared with those reported by Proffit et al. [25] (0.1–1 N), Moga et al. [10,11] (0.5 N), and Hemanth et al. [31,32] (0.3–1 N). Hofmann et al. [36,37], in two studies using the same HP failure criteria for the PDL, reported unusual qualitative and quantitative results for 0.5–1 N of intrusion, which contradicted his earlier study and the existing clinical knowledge. Maravic et al. [26], despite using a single FEA model, supplemented the numerical analysis with an in vivo–in vitro study, reporting numerical results that did not accurately match the experimental ones. 

The common issue found for all the FEA studies [19,20,21,26,28,29,31,32,33,34,36,37,38,39,40] was the small sample size of only one (one patient, a single 3D model, subjected to few orthodontic movements, generally in intact periodontium, and aleatory, using one or two failure criteria). This study and our earlier [2,10,11,12] studies tried to address these issues by increasing the sample size to nine, using multiple various bone loss levels, and rationally selecting the failure criteria (nine patients, eighty-one models and 405 numerical simulations), thus obtaining different and more accurate results. 

There are few studies (i.e., limited to PDL investigations) that approach the study of biomechanical behavior of dental tissues during the periodontal breakdown, while all numerical FEA examinations are of the intact periodontium. Moreover, there is no standardization (due to the lack of biomechanical engineering knowledge) for conducting finite element studies in the dentistry field [1]. Thus, there is a need for studies to supply data that address this issue, since numerical simulations are the only possible method to study living tissues.

The boundary conditions assumed in the FEA studies for dental tissues were homogeneity, isotropy, and linear elasticity [1,2,5,8,10,11,12,13,14,15,16,17,18,19,20,21,22,26,27,28,29,30,31,32,33,34,35,36,37,38,39,40], despite the acknowledged anisotropy, non-homogeneity, and nonlinear elasticity of dental tissues. However, in order for all the above assumptions to supply accurate results, some issues must be addressed. If the applied loads are around 1 N, small movements and displacements are produced, and all tissues display biomechanical linear–elastic isotropic behavior despite their non-linear anisotropic behavior under higher loads [2,10,11,12]. Few studies have addressed the differences between linear and non-linear behavior (and limited only to PDL behavior). Hemanth et al. [31,32] investigated these differences under light forces up to 1 N (intrusion and tipping of an upper incisor) and reported up to 20–50% less quantitative applied force needed for non-linearity vs. linearity. However, these two studies [31,32] did not address the essential issue of the adequate material-type failure criteria for PDL, since they used for comparison the S1 tensile and S3 compressive brittle criteria for the analysis of a ductile behavior resemblance material such as PDL. Another potential issue that could influence the results seems to be the accuracy of the analyzed 3D model (i.e., only 148,097 elements and 239,666 nodes and idealized anatomy) when extremely fine, small, and sensitive interactions occur both within and between anatomical elements.

The issues of assuming non-homogeneity in dental tissues demand an extremely complicated mathematical equations approach, internal micro-architecture tissue 3D modeling, and high computing resources, which simply are not possible to be performed manually since this type of modeling software is not available. However, in the engineering field, this problem has been approached and addressed by mathematically designing failure criteria for both ductile non-homogenous materials (T) and homogenous (VM) materials, thus supplying the only practical solution currently possible.

The anatomical accuracy of the models influences the correctness of the results, especially if small tissues under small displacements are investigated. Due to difficulties related to the creation of anatomically accurate 3D models (possible through a manual segmentation technique), and since the automated software detection algorithm encounters difficulties in identifying the differences of various shades of gray on the CBTC slices, most numerical analyses have a reduced sample size that is limited to only one model or use idealized models that do not accurately follow the anatomical accuracy. However, the downside of manual segmentation is the presence of a limited number of surface anomalies and irregularities (as in the models herein) that usually do not influence the accuracy of the results since these are in non-essential areas and the internal software mesh-testing algorithm allowed the simulations. 

Due to all these limitative issues, and since FEA analysis (here included) cannot entirely reproduce the clinical situation and biomechanical behavior, correlations with clinical behavioral biomechanics and physiological constants (such as MHP) are mandatory for confirming the numerical results. Thus, there is a need to have a multidisciplinary engineering and medical/dental knowledge approach that discusses all the above-mentioned issues, our team’s work herein and in earlier [2,10,11,12] studies being the first in this direction. Nevertheless, despite all the above limitative issues, FEA is still the only available method to study the stress and strain distribution in the tissue components of an anatomical tissue.

## 5. Practical/Clinical Implications

Periodontal breakdown is found in orthodontic patients, with little available information about the stress distribution under light orthodontic forces (0.5 N). The clinician needs to know not only the areas most affected by orthodontic stresses but also the amount of stress that could appear during the most stressful orthodontic movements to be able to individualize the treatment and correlate it with bone loss levels. Another practical issue is related to the fact that, based on the results herein, a single unitary suitable failure criterion for the study of teeth and the surrounding periodontium can be seen, which is strictly correlated with the physical properties and ductile resemblance of dental tissues. The researcher benefits from the biomechanical correlations herein, which supply important data for further studies that are needed for a better understanding of the biomechanical behavior of periodontal breakdown and for minimalizing the risks of further tissue loss.

## 6. Conclusions

For the numerical analysis of bone, the ductile failure criteria are suitable (both T and VM are adequate for the study of bone), with Tresca being more adequate than VM.

The S1, S3, and HP failure criteria, due to their distinctive design dedicated to brittle materials and liquids/gas, only occasionally correctly described the distribution of stresses in the bone.

For all five orthodontic movements and bone levels, only T and VM displayed a coherent and correlated gradual stress increase pattern along with the periodontal breakdown.

The quantitative values provided by T and VM were the highest among all five criteria (for each movement and level of bone loss).

The MHP (maximum physiological hydrostatic pressure) was exceeded in all simulations, since the mandibular bone is anatomically less vascularized, and the ischemic risks are reduced.

T and VM displayed rotation and translation, closely followed by tipping, as stressful movements, while intrusion and extrusion were less stressful for the mandibular bone.

Based on correlations with other earlier numerical studies on the same models and boundary conditions, T can be seen as a suitable single unitary failure criterion for the study of teeth and the surrounding periodontium.

## Figures and Tables

**Figure 1 medicina-59-01462-f001:**
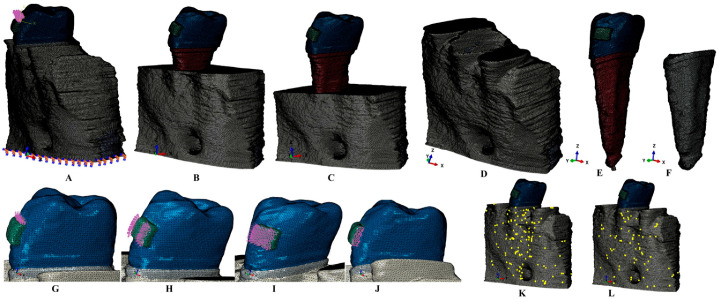
(**A**) 2nd lower right premolar model with intact periodontium, and applied vectors (encastered model base and extrusion loads); (**B**) 4 mm bone loss; (**C**) 8 mm bone loss; (**D**) bone structure (with cortical and trabecular components); (**E**) tooth model with bracket, enamel, dentin and neuro-vascular bundle, (**F**) intact PDL; applied vectors: (**G**) intrusion, (**H**) rotation, (**I**) tipping, (**J**) translation; (**K**) element warnings of the cortical bone component; (**L**) elements warnings of the cortical component.

**Figure 2 medicina-59-01462-f002:**
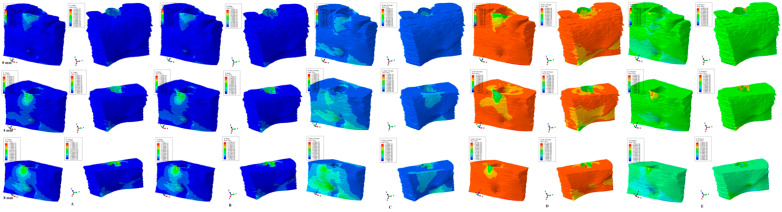
Comparative stress display of the five failure criteria in intact, 4 mm, and 8 mm periodontal breakdown for the extrusion movement under 0.5 N of load—vestibular and lingual views: (**A**) Tresca; (**B**) Von Mises; (**C**) maximum principal S1; (**D**) minimum principal S3; (**E**) pressure.

**Figure 3 medicina-59-01462-f003:**
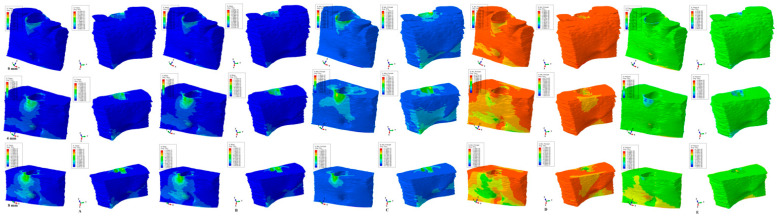
Comparative stress display of the five failure criteria in intact, 4 mm, and 8 mm periodontal breakdown for the intrusion movement under 0.5 N of load—vestibular and lingual views: (**A**) Tresca; (**B**) Von Mises; (**C**) maximum principal S1; (**D**) minimum principal S3; (**E**) pressure.

**Figure 4 medicina-59-01462-f004:**
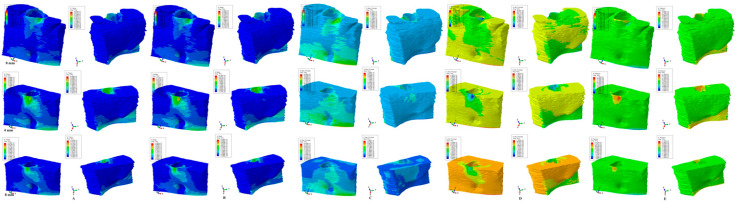
Comparative stress display of the five failure criteria in intact, 4 mm, and 8 mm periodontal breakdown for the rotation movement under 0.5 N of load—vestibular and lingual views: (**A**) Tresca; (**B**) Von Mises; (**C**) maximum principal S1; (**D**) minimum principal S3; (**E**) pressure.

**Figure 5 medicina-59-01462-f005:**
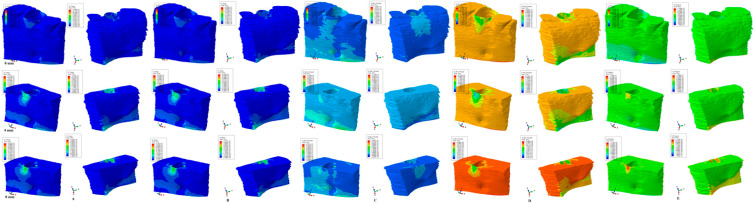
Comparative stress display of the five failure criteria in intact, 4 mm, and 8 mm periodontal breakdown for the tipping movement under 0.5 N of load—vestibular and lingual views: (**A**) Tresca; (**B**) Von Mises; (**C**) maximum principal S1; (**D**) minimum principal S3; (**E**) pressure.

**Figure 6 medicina-59-01462-f006:**
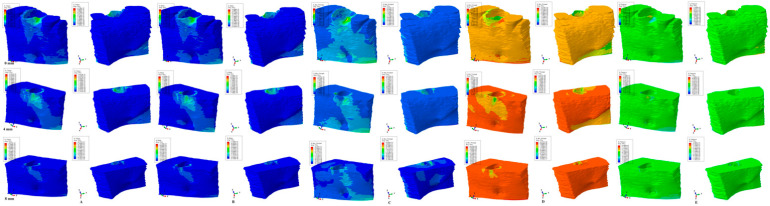
Comparative stress display of the five failure criteria in intact, 4 mm, and 8 mm periodontal breakdown for the translation movement under 0.5 N of load—vestibular and lingual views: (**A**) Tresca; (**B**) Von Mises; (**C**) maximum principal S1; (**D**) minimum principal S3; (**E**) pressure.

**Table 1 medicina-59-01462-t001:** Elastic properties of materials.

Material	Young’s Modulus, E (GPa)	Poisson Ratio, ʋ	Refs.
Enamel	80	0.33	[2,10,11,12]
Dentin/Cementum	18.6	0.31	[2,10,11,12]
Pulp	0.0021	0.45	[2,10,11,12]
PDL	0.0667	0.49	[2,10,11,12]
Cortical bone	14.5	0.323	[2,10,11,12]
Trabecular bone	1.37	0.3	[2,10,11,12]
Bracket (Stainless Steel)	190	0.265	[2,10,11,12]

**Table 2 medicina-59-01462-t002:** Maximum stress average values (KPa) produced by 0.5 N of orthodontic forces.

Resorption (mm)			0	1	2	3	4	5	6	7	8
Intrusion	**Tresca**	a	31.22	36.19	41.17	46.13	51.11	52.35	53.59	54.83	56.08
		m	31.22	31.93	32.64	33.35	34.07	36.06	38.06	40.00	42.06
		c	124.86	127.71	130.56	133.41	136.26	137.26	138.27	139.27	140.28
	**VM**	a	27.10	31.39	35.68	39.97	44.27	46.03	47.79	49.55	51.31
		m	27.10	27.70	28.31	28.91	29.52	31.76	34.00	36.24	38.48
		c	108.34	110.76	113.19	115.61	118.04	120.59	123.15	125.71	128.27
	P	a	21.94	22.62	23.29	23.96	24.64	25.04	25.45	25.85	26.26
		m	21.94	22.62	23.29	23.96	24.64	25.04	25.45	25.85	26.26
		c	−59.87	−56.71	−53.56	−50.40	−47.25	−44.69	−42.14	−39.58	−37.03
	S1	a	−6.33	−5.51	−4.69	3.87	3.05	3.55	−4.07	−4.57	−5.08
		m	−6.33	−5.51	−4.69	3.87	3.05	5.03	7.01	8.99	10.96
		c	155.22	145.32	135.42	125.52	115.62	121.54	127.47	133.40	139.33
	S3	a	−18.95	−22.68	−26.41	−30.14	−33.88	−33.92	−33.96	−33.96	−34.05
		m	−18.95	−22.68	−26.41	−30.14	−33.88	−33.92	−33.96	−33.96	−34.05
		c	−39.86	−38.36	−36.87	−35.37	−33.88	−33.92	−33.96	−33.96	−34.05
Extrusion	**Tresca**	a	31.22	36.19	41.17	46.13	51.11	52.35	53.59	54.83	56.08
		m	31.22	31.93	32.64	33.35	34.07	36.06	38.06	40.00	42.06
		c	124.86	127.71	130.56	133.41	136.26	137.26	138.27	139.27	140.28
	**VM**	a	27.10	31.39	35.68	39.97	44.27	46.03	47.79	49.55	51.31
		m	27.10	27.70	28.31	28.91	29.52	31.76	34.00	36.24	38.48
		c	108.34	110.76	113.19	115.61	118.04	120.59	123.15	125.71	128.27
	P	a	−12.84	−13.38	−13.93	−14.48	−16.65	−19.05	−21.45	−23.85	−26.26
		m	−12.84	−13.38	−13.93	−14.48	−16.65	−19.05	−21.45	−23.85	−26.26
		c	59.87	56.71	53.56	50.40	47.25	44.69	42.14	39.58	37.02
	S1	a	18.90	26.24	33.59	40.93	48.28	49.88	51.49	53.10	54.71
		m	18.90	26.24	33.59	40.93	48.28	49.88	51.49	53.10	54.71
		c	39.87	41.97	44.07	46.18	48.28	49.88	51.49	53.10	54.71
	S3	a	6.33	5.51	4.69	3.87	−3.05	3.55	4.07	4.57	5.08
		m	−9.83	−8.13	−6.44	−4.74	−3.05	−5.03	−7.01	−8.99	−10.97
		c	−155.22	−145.32	−135.42	−125.52	−115.62	−113.52	−111.43	−109.33	−107.24
Translation	**Tresca**	a	61.81	71.46	81.11	90.77	100.42	108.97	117.53	126.09	134.64
		m	61.81	71.46	81.11	90.77	100.42	108.97	117.53	126.09	134.64
		c	154.40	159.73	165.07	170.41	175.75	184.09	192.45	200.79	209.14
	**VM**	a	54.31	62.61	70.92	79.23	87.54	94.94	102.34	109.74	117.14
		m	54.31	62.61	70.92	79.23	87.54	94.94	102.34	109.74	117.14
		c	135.75	140.11	144.48	148.84	153.21	158.85	164.49	170.13	175.78
	P	a	−46.18	−42.14	−38.10	−34.01	−30.02	−26.79	−23.56	−20.33	−17.11
		m	−46.18	−42.14	−38.10	−34.01	−30.02	−26.79	−23.56	−20.33	−17.11
		c	−83.38	−78.39	−73.40	−68.41	−63.42	−59.90	−56.39	−52.88	−49.37
	S1	a	61.60	50.33	39.06	27.79	16.52	27.95	39.39	50.83	62.27
		m	36.62	31.59	26.57	21.54	16.52	27.95	39.39	50.83	62.27
		c	211.59	194.17	176.75	159.33	141.91	136.88	131.86	126.84	121.84
	S3	a	1.13	3.72	6.31	−8.90	−11.50	−10.03	−8.57	7.09	5.62
		m	1.13	3.72	6.31	−8.90	−11.50	−10.03	−8.57	7.09	5.62
		c	−161.14	−140.58	−120.03	−99.48	−78.93	−85.04	−91.16	−97.27	−103.39
Rotation	**Tresca**	a	63.95	70.54	77.14	83.73	90.33	99.18	108.03	116.88	125.74
		m	63.95	70.54	77.14	83.73	90.33	99.18	108.03	116.88	125.74
		c	255.74	268.98	282.22	295.46	308.71	315.36	322.02	328.68	335.34
	**VM**	a	55.47	61.21	66.95	72.69	78.43	86.08	91.82	99.47	109.03
		m	55.47	61.21	66.95	72.69	78.43	86.08	91.82	99.47	109.03
		c	221.83	231.67	241.52	251.37	261.22	268.59	275.97	283.34	290.72
	P	a	20.61	21.27	21.94	22.60	23.27	27.44	31.61	35.78	39.95
		m	20.61	21.27	21.94	22.60	23.27	27.44	31.61	35.78	39.95
		c	77.87	82.23	86.59	90.95	95.32	96.17	97.03	97.88	98.74
	S1	a	43.95	53.70	63.45	73.20	82.96	89.68	96.40	103.12	109.84
		m	43.95	46.01	48.07	50.13	52.20	57.90	63.60	69.30	75.00
		c	161.35	134.06	106.78	79.48	52.20	57.90	63.60	69.30	75.00
	S3	a	−5.09	−6.79	−8.49	10.19	11.89	13.71	15.53	−17.35	−19.17
		m	−95.49	−108.74	121.99	−135.24	−148.50	−167.16	−185.83	−204.49	−223.16
		c	−185.93	−208.66	−231.39	−254.12	−276.86	−273.63	−270.41	−267.18	−263.96
Tipping	**Tresca**	a	61.06	65.40	69.75	74.10	78.45	81.51	84.58	87.64	90.71
		m	61.06	65.40	69.75	74.10	78.45	81.51	84.58	87.64	90.71
		c	122.11	130.81	139.51	148.21	156.91	163.03	169.16	175.29	181.41
	**VM**	a	53.07	56.78	60.50	64.22	67.94	70.74	73.54	76.34	79.14
		m	53.07	56.78	60.50	64.22	67.94	70.74	73.54	76.34	79.14
		c	106.18	113.58	120.99	128.40	135.81	141.44	147.08	152.72	158.36
	P	a	12.94	14.82	16.70	18.58	20.46	25.31	30.16	35.01	39.87
		m	12.94	14.82	16.70	18.58	20.46	25.31	30.16	35.01	39.87
		c	52.74	62.11	71.49	80.86	90.24	96.93	103.63	110.33	117.03
	S1	a	10.18	16.86	23.54	30.22	36.90	44.14	51.38	58.62	65.86
		m	10.18	16.86	23.54	30.22	36.90	44.14	51.38	58.62	65.86
		c	53.18	49.11	45.04	40.97	36.90	44.14	51.38	58.62	65.86
	S3	a	8.52	10.07	11.63	13.19	14.72	16.01	17.30	18.59	19.89
		m	8.52	9.27	10.03	−10.78	−11.54	−14.43	−17.33	−20.22	−23.12
		c	−171.86	−184.27	−196.69	−209.11	−221.53	−236.44	−251.35	−266.26	−281.18

a—apical third; m—middle third; c—cervical third.

## Data Availability

Not applicable.

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
