# Peer review of "Cortical and Trabecular Bone Stress Assessment during Periodontal Breakdown–A Comparative Finite Element Analysis of Multiple Failure Criteria"

_medicina, 2023, doi:10.3390/medicina59081462_

Round 1

Reviewer 1 Report

“Cortical and Trabecular Bone Stress Assessment During Periodontal Breakdown- a comparative analysis of multiple failure criteria” was submitted to Medicina.

The manuscript deals with an interesting issue; however, there are several concerns related to the study.

Title: It should be noted that this study uses FEA

Abstract

The objectives described do not coincide with those of the introduction.

Line 23. Define T.

Line 24. Define HP.

Lines 25-26. “coherent and correlated gradual stress increase pattern” is very subjective information. Is it possible to provide data or more tangible information?

Lines 26-27. “T and VM were the highest”. Could you quantify the value? For example, double, triple, or statistically? Please consider these recommendations in the results described in lines 28 and 29, and in the results described in the main text.

Line 27. Define MHP.

Line 29. Conclusions related to objective "a" described in the introduction should be presented.

On line 23 you mention "intrusion, extrusion, tipping, rotation, and translation". Considering the relevance of these movements to clinicians, describe the conclusions related to such movements.

Keywords: Terms should be reviewed. Please verify that they are MeSH terms.

Introduction

The introduction should be rewritten and described in a more orderly way. There are topics that are treated in different paragraphs and many ideas are repeated. It is recommended to consolidate in each paragraph the concepts related to each of the topics covered in an orderly and coherent way that relates the paragraphs.

Line 38. Many references (8) are presented for a single concept.

Line 69. Many references are presented. It is recommended that the most up-to-date ones be kept. Some are over 10 years old. Consider this recommendation throughout the manuscript.

Many acronyms are used. It is recommended to leave only the most necessary. Everyone's definition should be reviewed; for example, NVB is not defined.

M&M

Line 183. It must be indicated that it is a convenience sample. Ideally, a sample size calculation should be performed.

Lines 185-189. Were there exclusion criteria?

Results

Lines 248-249. Please remember that not all readers are experts in these assessments. Please explain how the lack of visible influences is corroborated.

Consider the recommendations given in the abstract about the results.

Discussion

It is recommended to carry out the discussion keeping the same numerals described in the results.

The limitations of FEA are described. The limitations of the present study must also be present.

The clinical implications should be presented at the end of the discussion.

Conclusions

Relevant conclusions that are not described in the abstract are presented. Remember that the abstract has a capacity of 300 words.

moderate editing of English language required

Author Response

Department of Cariology, Endodontics and Oral Pathology

School of Dental Medicine

University of Medicine and Pharmacy

Ms. Amelia Men

Assistant Editor

Medicina        

Special Issue - Dentistry: A Multidisciplinary Approach  

                                                                                                                                 August 6th, 2023

Dear Ms. Amelia Men,

Thank you very much for your letter dated August 5th, 2023, with the comments of the reviewers. We have now carefully considered the comments of the reviewers and amended the paper accordingly. All changes are highlighted in red throughout the manuscript and included also below.

Reply to Reviewer #1:

We agree and we thank the reviewer for his/her time and comments. Appropriate changes in the manuscript have by now been made. Please see below and in the manuscript.

Concern of the reviewer:

” Comments and Suggestions for Authors

“Cortical and Trabecular Bone Stress Assessment During Periodontal Breakdown- a comparative analysis of multiple failure criteria” was submitted to Medicina.

The manuscript deals with an interesting issue; however, there are several concerns related to the study.

Title: It should be noted that this study uses FEA

Abstract

The objectives described do not coincide with those of the introduction.

Line 23. Define T.

Line 24. Define HP.

Lines 25-26. “coherent and correlated gradual stress increase pattern” is very subjective information. Is it possible to provide data or more tangible information?

Lines 26-27. “T and VM were the highest”. Could you quantify the value? For example, double, triple, or statistically? Please consider these recommendations in the results described in lines 28 and 29, and in the results described in the main text.

Line 27. Define MHP.

Line 29. Conclusions related to objective "a" described in the introduction should be presented.

On line 23 you mention "intrusion, extrusion, tipping, rotation, and translation". Considering the relevance of these movements to clinicians, describe the conclusions related to such movements.

Keywords: Terms should be reviewed. Please verify that they are MeSH terms.

Introduction

The introduction should be rewritten and described in a more orderly way. There are topics that are treated in different paragraphs and many ideas are repeated. It is recommended to consolidate in each paragraph the concepts related to each of the topics covered in an orderly and coherent way that relates the paragraphs.

Line 38. Many references (8) are presented for a single concept.

Line 69. Many references are presented. It is recommended that the most up-to-date ones be kept. Some are over 10 years old. Consider this recommendation throughout the manuscript.

Many acronyms are used. It is recommended to leave only the most necessary. Everyone's definition should be reviewed; for example, NVB is not defined.

M&M

Line 183. It must be indicated that it is a convenience sample. Ideally, a sample size calculation should be performed.

Lines 185-189. Were there exclusion criteria?

Results

Lines 248-249. Please remember that not all readers are experts in these assessments. Please explain how the lack of visible influences is corroborated.

Consider the recommendations given in the abstract about the results.

Discussion

It is recommended to carry out the discussion keeping the same numerals described in the results.

The limitations of FEA are described. The limitations of the present study must also be present.

The clinical implications should be presented at the end of the discussion.

Conclusions

Relevant conclusions that are not described in the abstract are presented. Remember that the abstract has a capacity of 300 words.

Comments on the Quality of English Language

moderate editing of English language required.”

Point-by-point response to the reviewer’s comments:

  1. Concern of the reviewer:

“Title: It should be noted that this study uses FEA”

Our response:

  • We thank the reviewer for his/her concern and comments. We do hope that our changes are according to the reviewer‘s remarks.

Revised text: pg.1 line 3

“Cortical and Trabecular Bone Stress Assessment During Periodontal Breakdown – a FEA comparative analysis of multiple failure criteria.”

  1. Concern of the reviewer:

“Abstract

  1. The objectives described do not coincide with those of the introduction.

            Line 23. Define T.

            Line 24. Define HP.

  1. Lines 25-26. “coherent and correlated gradual stress increase pattern” is very subjective information. Is it possible to provide data or more tangible information?
  2. Lines 26-27. “T and VM were the highest”. Could you quantify the value? For example, double, triple, or statistically? Please consider these recommendations in the results described in lines 28 and 29, and in the results described in the main text.
  3. Line 27. Define MHP.
  4. Line 29. Conclusions related to objective "a" described in the introduction should be presented.
  5. On line 23 you mention "intrusion, extrusion, tipping, rotation, and translation". Considering the relevance of these movements to clinicians, describe the conclusions related to such movements.”

Our response:

  • We thank the reviewer for his/her concern and comments. We do hope that our changes are according to the reviewer‘s remarks.
  1. Revised text: pg.1 lines 17-22

“This numerical analysis investigated the biomechanical behavior of the mandibular bone as structure subjected to 0.5 N of orthodontic force during periodontal breakdown. Additionally, assessed the suitability of the five most used failure criteria [Von Mises (VM), Tresca (T), Maximum Principal (S1), Minimum Principal (S3), and Hydrostatic Pressure (HP)] for the study of bone and identify a single criterion for the study of tooth and surrounding periodontium (by performing correlations with other FEA studies).” 

  1. Revised text: pg.1 lines 28-29

“Only T and VM displayed a coherent and correlated gradual stress increase pattern along for all five movements and levels of the periodontal breakdown.”

  1. Revised text: pg.1 lines 29-30

“The quantitative values provided by T and VM were the highest (for each movement and level of bone loss) among all five criteria.”

Since there are three points of investigation of bone stress (apical third, middle third, cervical third) for each movement, level of bone loss and failure criteria (please see Table 2), we were unable to find a more precise expression of the results. However, this expression fully sustains the results and conclusions of the manuscript.

  1. Revised text: pg.1 lines 31-32

“MHP (maximum physiological hydrostatic pressure) was exceeded in all simulations since the mandibular bone is anatomically less vascularized, and the ischemic risks are reduced.”

  1. Revised text: pg.1 lines 24-37

“For the numerical analysis of bone, the ductile failure criteria are suitable (both T and VM adequate for the study of bone), with Tresca more suited. S1, S3 and HP criteria due to their distinctive design dedicated to brittle materials and liquids/gas only occasionally correctly described the bone stress distribution. Only T and VM displayed a coherent and correlated gradual stress increase pattern for all five movements and levels of the periodontal breakdown. The quantitative values provided by T and VM were the highest (for each movement and level of bone loss) among all five criteria. MHP (maximum physiological hydrostatic pressure) was exceeded in all simulations since the mandibular bone is anatomically less vascularized, and the ischemic risks are reduced. Only T and VM displayed a corelated (both qualitative and quantitative) stress increase for all five movements. Both T and VM displayed rotation and translation, closely followed by tipping as stressful movements, while intrusion and extrusion were less stressful for the mandibular bone. Based on correlations with earlier numerical studies on the same models and boundary conditions, T seems better suited as single unitary failure criterion for the study of tooth and surrounding periodontium.”

  1. Revised text: pg.1 lines 32-37

“Only T and VM displayed a corelated (both qualitative and quantitative) stress increase for all five movements. Both T and VM displayed rotation and translation, closely followed by tipping as stressful movements, while intrusion and extrusion were less stressful for the mandibular bone. Based on correlations with earlier numerical studies on the same models and boundary conditions, T seems better suited as single unitary failure criterion for the study of tooth and surrounding periodontium”

  1. Concern of the reviewer:

“Keywords: Terms should be reviewed. Please verify that they are MeSH terms.”

Our response:

  • We thank the reviewer for his/her concern and comments. We do hope that our changes are according to the reviewer‘s remarks.

    Revised text: pg.1 lines 38-39

“Bone; bone loss; orthodontic force; Finite Element Analysis; orthodontic movement.”

  1. Concern of the reviewer:
  2. “-The introduction should be rewritten and described in a more orderly way. There are topics that are treated in different paragraphs and many ideas are repeated. It is recommended to consolidate in each paragraph the concepts related to each of the topics covered in an orderly and coherent way that relates the paragraphs.

           Line 38. Many references (8) are presented for a single concept.

           Line 69. Many references are presented. It is recommended that the most up-to-date ones    

            be kept. Some are over 10 years old. Consider this recommendation throughout the

             manuscript.

  1. Many acronyms are used. It is recommended to leave only the most necessary. Everyone's definition should be reviewed; for example, NVB is not defined.”

Our response:

  • We thank the reviewer for his/her concern and comments. We do hope that our changes are according to the reviewer‘s remarks.
  1. This manuscript is the first research of this type assessing the biomechanical behavior of bone structure during periodontal breakdown. Moreover, no other study employed multiple failure criteria for the finite element analysis of bone in the search of a single most accurate criteria for the study of bone and dental tissues (tooth and surrounding periodontium included). There are multiple concepts (with correlations and relationships) approached in the introduction section that needed to be presented for familiarizing the reader (both clinician and researcher) for a better understanding of the study. Regarding the old references, it must be emphasized that there are only few data available in the research flow regarding herein subject, so for a better support of the explanations we needed to use whatever resources were available.

  1. Revised text:3 lines 134-137

 “For avoiding ischemia, necrosis, and further periodontal loss the physiological maximum hydrostatic pressure of 16 KPa [2, 10-12] should not be exceeded, especially in the well vascularized and easily deformable under stresses dental tissues (i.e., PDL, dental pulp and neuro-vascular bundle [NVB]).”

All the acronyms displayed in the herein manuscript were properly defined, keeping only the most necessary for an easier reading and understanding of the text. 

  1. Concern of the reviewer:

“M&M

  1. Line 183. It must be indicated that it is a convenience sample. Ideally, a sample size calculation should be performed.
  2. Lines 185-189. Were there exclusion criteria?.”

Our response:

  • We thank the reviewer for his/her concern and comments. We do hope that our changes are according to the reviewer‘s remarks.
  1. Revised text:4 lines 192-195

“The selected convenience sample size of nine was acceptable for the accuracy of the results since most of the previous FEA studies employed a sample size of one (one patient, one 3D model and few simulations) [1, 2, 5, 8, 10-22, 26-40].”

  1. Revised text:4 lines 198-200

“All the situations that were not covered by the above criteria were considered to be exclusion criteria (especially the lack of arch integrity, tooth malposition, more than 8 mm bone loss cases and inflamed periodontium).”

  1. Concern of the reviewer:

“Results

Lines 248-249. Please remember that not all readers are experts in these assessments. Please explain how the lack of visible influences is corroborated.

Consider the recommendations given in the abstract about the results.”

Our response:

  • We thank the reviewer for his/her concern and comments. We do hope that our changes are according to the reviewer‘s remarks.

Revised text: pg.6 lines 258-260

“Our analysis involved eighty-one mandibular models in 405 FEA numerical simulations (Figures 2-6 and Table 2). No influences due to age, sex, or periodontal status were seen.”

  1. Concern of the reviewer:

“Discussion

  1. It is recommended to carry out the discussion keeping the same numerals described in the results.
  2. The limitations of FEA are described. The limitations of the present study must also be present.
  3. The clinical implications should be presented at the end of the discussion.”

Our response:

  • We thank the reviewer for his/her concern and comments. We do hope that our changes are according to the reviewer‘s remarks.
  1. Revised text:12 lines 400-404

“During the periodontal breakdown simulations under the five orthodontic movements only T (Tresca) and VM (Von Mises) criteria displayed both qualitatively and quantitatively a coherent and correlated pattern, and a biomechanical behavior resembling in vivo (Figures 2-6 and Table 2). The other three criteria (S1- maximum tensile, S3 – minimum compressive, and HP - hydrostatic pressure) displayed no visible biomechanical… .”

  1. Revised text:15 lines 598-599

“Due to all these limitative issues and that FEA analysis (here included) cannot entirely reproduce clinical situation and biomechanical behavior… .”

  1. Revised text:16 lines 606-617
  2. Concern of the reviewer:

“Conclusions

Relevant conclusions that are not described in the abstract are presented. Remember that the abstract has a capacity of 300 words.”

Our response:

  • We thank the reviewer for his/her concern and comments. We do hope that our changes are according to the reviewer‘s remarks.

Revised text: pg.1 lines 17-37

Abstract: This numerical analysis investigated the biomechanical behavior of the mandibular bone as structure subjected to 0.5 N of orthodontic force during periodontal breakdown. Additionally, assessed the suitability of the five most used failure criteria [Von Mises (VM), Tresca (T), Maximum Principal (S1), Minimum Principal (S3), and Hydrostatic Pressure (HP)] for the study of bone and identify a single criterion for the study of tooth and surrounding periodontium (by performing correlations with other FEA studies). The finite elements analysis (FEA) employed 405 simulations over eighty-one mandibular models with variable levels of bone loss (0-8 mm) and five orthodontic movements (intrusion, extrusion, tipping, rotation, and translation). For the numerical analysis of bone, the ductile failure criteria are suitable (both T and VM adequate for the study of bone), with Tresca more suited. S1, S3 and HP criteria due to their distinctive design dedicated to brittle materials and liquids/gas only occasionally correctly described the bone stress distribution. Only T and VM displayed a coherent and correlated gradual stress increase pattern for all five movements and levels of the periodontal breakdown. The quantitative values provided by T and VM were the highest (for each movement and level of bone loss) among all five criteria. MHP (maximum physiological hydrostatic pressure) was exceeded in all simulations since the mandibular bone is anatomically less vascularized, and the ischemic risks are reduced. Only T and VM displayed a corelated (both qualitative and quantitative) stress increase for all five movements. Both T and VM displayed rotation and translation, closely followed by tipping as stressful movements, while intrusion and extrusion were less stressful for the mandibular bone. Based on correlations with earlier numerical studies on the same models and boundary conditions, T seems better suited as single unitary failure criterion for the study of tooth and surrounding periodontium.”

  1. Concern of the reviewer:

“Comments on the Quality of English Language moderate editing of English language required.”

Our response:

  • We thank the reviewer for his/her concern and comments. We do hope that our changes are according to the reviewer‘s remarks.

Revised text: entire manuscript

Reviewer 2 Report

Dear Authors,

After reading and reviewing your manuscript, I would like to appreciate your team for the work.

Although the manuscript is well written, I would like the authors to elaborate the information in a brief manner for the queries below, so the manuscript will be informative and interesting for the readers.

What are the variations between cortical and trabecular bone, and how do they affect jaw biomechanics during periodontal breakdown?

How does periodontal disease influence stress distribution in cortical and trabecular bone?

Is there any part of the jaw where cortical or trabecular bone stress is particularly noticeable during periodontal breakdown?

What imaging modalities are routinely used to evaluate cortical and trabecular bone stress in periodontal disease patients?

Can measuring cortical and trabecular bone stress help in the early identification and diagnosis of periodontal disease?

How do variations in cortical and trabecular bone thickness and density affect stress distribution during periodontal disease progression?

Are there any gender or age variations in cortical and trabecular function?

Are there any gender or age variations in cortical and trabecular bone stress patterns in periodontitis patients?

How do different periodontal disease treatments impact cortical and trabecular bone stress distribution?

What is the function of finite element analysis (FEA) in assessing cortical and trabecular bone stress during periodontal breakdown?

Are there any viable therapeutic treatments that can particularly target cortical and trabecular bone stress to enhance periodontal disease treatment outcomes?

Author Response

Department of Cariology, Endodontics and Oral Pathology

School of Dental Medicine

University of Medicine and Pharmacy

Ms. Amelia Men

Assistant Editor

Medicina        

Special Issue - Dentistry: A Multidisciplinary Approach  

                                                                                                                                 August 6th, 2023

Dear Ms. Amelia Men,

Thank you very much for your letter dated August 5th, 2023, with the comments of the reviewers. We have now carefully considered the comments of the reviewers and amended the paper accordingly. All changes are highlighted in red throughout the manuscript and included also below.

Reply to Reviewer #2

We agree and we thank the reviewer for his/her time and comments. Appropriate changes in the manuscript have by now been made. Please see below and in the manuscript.

Concern of the reviewer:

” Comments and Suggestions for Authors

Dear Authors,

After reading and reviewing your manuscript, I would like to appreciate your team for the work.

Although the manuscript is well written, I would like the authors to elaborate the information in a brief manner for the queries below, so the manuscript will be informative and interesting for the readers.

What are the variations between cortical and trabecular bone, and how do they affect jaw biomechanics during periodontal breakdown?

How does periodontal disease influence stress distribution in cortical and trabecular bone?

Is there any part of the jaw where cortical or trabecular bone stress is particularly noticeable during periodontal breakdown?

What imaging modalities are routinely used to evaluate cortical and trabecular bone stress in periodontal disease patients?

Can measuring cortical and trabecular bone stress help in the early identification and diagnosis of periodontal disease?

How do variations in cortical and trabecular bone thickness and density affect stress distribution during periodontal disease progression?

Are there any gender or age variations in cortical and trabecular function?

Are there any gender or age variations in cortical and trabecular bone stress patterns in periodontitis patients?

How do different periodontal disease treatments impact cortical and trabecular bone stress distribution?

What is the function of finite element analysis (FEA) in assessing cortical and trabecular bone stress during periodontal breakdown?

Are there any viable therapeutic treatments that can particularly target cortical and trabecular bone stress to enhance periodontal disease treatment outcomes?.”

Point-by-point response to the reviewer’s comments:

  1. Concern of the reviewer:

“What are the variations between cortical and trabecular bone, and how do they affect jaw biomechanics during periodontal breakdown?”

Our response:

  • We thank the reviewer for his/her concern and comments. We do hope that our changes are according to the reviewer‘s remarks.

Revised text: pg.2 lines 51-62

“Bone as a continuum and single stand structure possess a high adaptation ability altering its geometry to provide the strongest structure possible with a minimum amount of tissue [7]. The bone structure also could absorb and dissipate the energy/stresses and elastically deform preventing the facture and/or destruction [7]. Internal structural micro-architecture allows microcracks/damage (i.e., linear, and diffuse microcracks, and microfractures) and time to heal [7]. Linear microcracks appear as response to compressive stresses (older age, more brittleness), the diffuse microdamage as response to tensile stresses (younger age, more ductileness), while microfractures as response to shear stresses (older age, mixt ductile brittleness, mostly in trabecular bone) [7].”

The bone acts like a continuum and single stand structure both in intact and during the bone loss that supports and absorb the loads.

  1. Concern of the reviewer:

“How does periodontal disease influence stress distribution in cortical and trabecular bone?.”

Our response:

  • We thank the reviewer for his/her concern and comments. We do hope that our changes are according to the reviewer‘s remarks.

Revised text: pg.1 lines 24-37

“For the numerical analysis of bone, the ductile failure criteria are suitable (both T and VM adequate for the study of bone), with Tresca more suited. S1, S3 and HP criteria due to their distinctive design dedicated to brittle materials and liquids/gas only occasionally correctly described the bone stress distribution. Only T and VM displayed a coherent and correlated gradual stress increase pattern for all five movements and levels of the periodontal breakdown. The quantitative values provided by T and VM were the highest (for each movement and level of bone loss) among all five criteria. MHP (maximum physiological hydrostatic pressure) was exceeded in all simulations since the mandibular bone is anatomically less vascularized, and the ischemic risks are reduced. Only T and VM displayed a corelated (both qualitative and quantitative) stress increase for all five movements. Both T and VM displayed rotation and translation, closely followed by tipping as stressful movements, while intrusion and extrusion were less stressful for the mandibular bone. Based on correlations with earlier numerical studies on the same models and boundary conditions, T seems better suited as single unitary failure criterion for the study of tooth and surrounding periodontium.” 

The bone acts like a continuum and single stand structure with support and absorption-dissipation role, so the bone loss produces both an increase of the amount of quantitative stress and a qualitative shift of stress distributing areas.

  1. Concern of the reviewer:

“Is there any part of the jaw where cortical or trabecular bone stress is particularly noticeable during periodontal breakdown?.”

Our response:

  • We thank the reviewer for his/her concern and comments. We do hope that our changes are according to the reviewer‘s remarks.

Revised text: pg.6 lines 270-289

“From the quantitative point of view (Table 2), the unusual biomechanical behavior was seen in S1 and S3 criteria in the rotation, translation, and tipping movements during the periodontal breakdown with a variable decrease/increase form 0-4 mm and 4-8 mm bone loss. The HP criteria displayed a decrease of amounts of stress for intrusion, extrusion and translation, and an increase for rotation and tipping. No stress pattern and/or correlations were observed between S1, S3, and HP criteria despite investigating the same models and in the same conditions as T and VM. Only T and VM criteria displayed a constant expected increase pattern during the periodontal breakdown process for all five movements confirming the acknowledged clinical biomechanical behavior. The rotation movement seemed to be the most bone stressful of all five criteria. Nevertheless, T and VM displayed rotation and translation, closely followed by tipping as stressful movements, while intrusion and extrusion were less stressful for the mandibular bone.

All quantitative stresses displayed in the mandibular bone for all five failure criteria exceeded the MHP of 16 KPa in both intact and 8 mm reduced periodontium. The highest amount of stress was produced by the rotation at 8 mm of bone loss (335KPa [0.335MPa]) under T criteria. The vestibular cervical third stress was the highest for all criteria, movements, and levels of bone loss. All stresses were lower than the acknowledged physical properties of bone: cortical bone compressive modulus - 16.7 GPa and compressive strength - 157 MPa; trabecular/cancellous bone compressive modulus - 0.155 GPa and compressive strength - 6 MPa [1, 5, 8, 13-18].”

Each type of movement and particular level of bone loss produces individual qualitative and quantitative results. The results can be better interpreted only using the correct failure criteria and in the general picture of the stress distribution in the entire complex structure tooth-surrounding periodontium.

  1. Concern of the reviewer:

“What imaging modalities are routinely used to evaluate cortical and trabecular bone stress in periodontal disease patients?.”

Our response:

  • We thank the reviewer for his/her concern and comments. We do hope that our changes are according to the reviewer‘s remarks.

Revised text: pg.2 lines 85-93

“The orthodontic biomechanical behavior of bone is influenced by the anatomy of tissues, materials, magnitude, the quantity and quality of the bone [1, 14]. There are several tools to analyze biomechanical behavior of bone and tooth, including in vitro (pho-to-elastic stress analysis, static/dynamic mechanical fracture tests) and numerical simulations (finite elements analysis) [1]. FEA is the only method that allows individual analysis of each component of a structure providing accurate results if the input data (anatomical accuracy and loading conditions) are correct [1, 13]. Only a numerical simulation as FEA allows correct biomechanical studies that assess and predict stress distribution in living dental tissues [13, 14, 17, 22].”

There are no current imaging modalities routinely used to evaluate the stress distribution. The FEA is a research method that cannot be clinically currently used, except in particular cases (since consumes a lot of time and resources).

  1. Concern of the reviewer:

“Can measuring cortical and trabecular bone stress help in the early identification and diagnosis of periodontal disease?”

Our response:

  • We thank the reviewer for his/her concern and comments. We do hope that our changes are according to the reviewer‘s remarks.

Revised text: pg.16 lines 606-617

“Periodontal breakdown is found in orthodontic patients, with little available information about the stress distribution under the light orthodontic forces (0.5 N). The clinician needs to know not only the most affected areas by orthodontic stresses but also the amount of stress that could appear during the most stressful orthodontic movements, to be able to individualize the treatment and correlate with bone loss levels. Another practical issue is related to the fact that based on herein results, a single unitary suitable failure criterion for the study of tooth and surrounding periodontium can be seen, strictly correlated with the physical properties and ductile resemblance of dental tissues. The researcher benefits from the herein biomechanical correlations, supplying important data for further studies that are needed for a better understanding of the biomechanical behavior of periodontal breakdown and minimalizing the risks of further tissue loss.”

The stress measurements cannot help an early identification and diagnosis of periodontal disease, since FEA is a research method that cannot be used in current clinical practice (i.e., time consuming). However, if a new software that automatically identifies the tissues on the CBCT images and combines them into a single model, and automatically performs the finite analyses, there are possibilities of clinical use.

  1. Concern of the reviewer:

“How do variations in cortical and trabecular bone thickness and density affect stress distribution during periodontal disease progression?”

Our response:

  • We thank the reviewer for his/her concern and comments. We do hope that our changes are according to the reviewer‘s remarks.

Revised text: pg.1 lines 24-37

“For the numerical analysis of bone, the ductile failure criteria are suitable (both T and VM adequate for the study of bone), with Tresca more suited. S1, S3 and HP criteria due to their distinctive design dedicated to brittle materials and liquids/gas only occasionally correctly described the bone stress distribution. Only T and VM displayed a coherent and correlated gradual stress increase pattern for all five movements and levels of the periodontal breakdown. The quantitative values provided by T and VM were the highest (for each movement and level of bone loss) among all five criteria. MHP (maximum physiological hydrostatic pressure) was exceeded in all simulations since the mandibular bone is anatomically less vascularized, and the ischemic risks are reduced. Only T and VM displayed a corelated (both qualitative and quantitative) stress increase for all five movements. Both T and VM displayed rotation and translation, closely followed by tipping as stressful movements, while intrusion and extrusion were less stressful for the mandibular bone. Based on correlations with earlier numerical studies on the same models and boundary conditions, T seems better suited as single unitary failure criterion for the study of tooth and surrounding periodontium.” 

The bone loss modifies the biomechanical behavior of tooth and surrounding periodontium, creating increased quantitative stress and qualitatively changing the stress areas, individually for each movement and level of bone loss.

  1. Concern of the reviewer:

“Are there any gender or age variations in cortical and trabecular function?”

Our response:

  • We thank the reviewer for his/her concern and comments. We do hope that our changes are according to the reviewer‘s remarks.

Revised text: pg.6 lines 258-260

“Our analysis involved eighty-one mandibular models in 405 FEA numerical simulations (Figures 2-6 and Table 2). No influences due to age, sex, or periodontal status were seen.”

No variation was seen.

  1. Concern of the reviewer:

“Are there any gender or age variations in cortical and trabecular bone stress patterns in periodontitis patients?”

Our response:

  • We thank the reviewer for his/her concern and comments. We do hope that our changes are according to the reviewer‘s remarks.

Revised text: pg.6 lines 258-260

“Our analysis involved eighty-one mandibular models in 405 FEA numerical simulations (Figures 2-6 and Table 2). No influences due to age, sex, or periodontal status were seen.”

No variation was seen.

  1. Concern of the reviewer:

“How do different periodontal disease treatments impact cortical and trabecular bone stress distribution?”

Our response:

  • We thank the reviewer for his/her concern and comments. We do hope that our changes are according to the reviewer‘s remarks.

Revised text: pg.4 lines 176-181

“The objectives of herein FEA analysis are: a. to assess the biomechanical behavior of mandibular bone subjected to light orthodontic forces during a horizontal periodontal breakdown; b. to assess the suitability for the study of bone of five of the most used failure criteria employed in dental tissues research; c. to correlate the results with other FEA related reports of dental tissues for identifying a single unitary suitable failure criteria for the study of tooth and surrounding periodontium.”

Our study focused on the above aims.

  1. Concern of the reviewer:

“What is the function of finite element analysis (FEA) in assessing cortical and trabecular bone stress during periodontal breakdown?”

Our response:

  • We thank the reviewer for his/her concern and comments. We do hope that our changes are according to the reviewer‘s remarks.

Revised text: pg.1 lines 17-37

Abstract: This numerical analysis investigated the biomechanical behavior of the mandibular bone as structure subjected to 0.5 N of orthodontic force during periodontal breakdown. Additionally, assessed the suitability of the five most used failure criteria [Von Mises (VM), Tresca (T), Maximum Principal (S1), Minimum Principal (S3), and Hydrostatic Pressure (HP)] for the study of bone and identify a single criterion for the study of tooth and surrounding periodontium (by performing correlations with other FEA studies). The finite elements analysis (FEA) employed 405 simulations over eighty-one mandibular models with variable levels of bone loss (0-8 mm) and five orthodontic movements (intrusion, extrusion, tipping, rotation, and translation). For the numerical analysis of bone, the ductile failure criteria are suitable (both T and VM adequate for the study of bone), with Tresca more suited. S1, S3 and HP criteria due to their distinctive design dedicated to brittle materials and liquids/gas only occasionally correctly described the bone stress distribution. Only T and VM displayed a coherent and correlated gradual stress increase pattern for all five movements and levels of the periodontal breakdown. The quantitative values provided by T and VM were the highest (for each movement and level of bone loss) among all five criteria. MHP (maximum physiological hydrostatic pressure) was exceeded in all simulations since the mandibular bone is anatomically less vascularized, and the ischemic risks are reduced. Only T and VM displayed a corelated (both qualitative and quantitative) stress increase for all five movements. Both T and VM displayed rotation and translation, closely followed by tipping as stressful movements, while intrusion and extrusion were less stressful for the mandibular bone. Based on correlations with earlier numerical studies on the same models and boundary conditions, T seems better suited as single unitary failure criterion for the study of tooth and surrounding periodontium.”

Revised text: pg.2 lines 89-93

“FEA is the only method that allows individual analysis of each component of a structure providing accurate results if the input data (anatomical accuracy and loading conditions) are correct [1, 13]. Only a numerical simulation as FEA allows correct biomechanical studies that assess and predict stress distribution in living dental tissues [13, 14, 17, 22].”

FEA is the only method that allows an individual image of stress distribution in each component of the tooth and periodontium, aiming to know how the stress pattern is influenced by the bone loss.

  1. Concern of the reviewer:

“Are there any viable therapeutic treatments that can particularly target cortical and trabecular bone stress to enhance periodontal disease treatment outcomes?”

Our response:

  • We thank the reviewer for his/her concern and comments. We do hope that our changes are according to the reviewer‘s remarks.

Revised text: pg.4 lines 176-181

“The objectives of herein FEA analysis are: a. to assess the biomechanical behavior of mandibular bone subjected to light orthodontic forces during a horizontal periodontal breakdown; b. to assess the suitability for the study of bone of five of the most used failure criteria employed in dental tissues research; c. to correlate the results with other FEA re-lated reports of dental tissues for identifying a single unitary suitable failure criteria for the study of tooth and surrounding periodontium.”

Our study focused on the above aims. However, the practical and clinicals implications of herein study are the following (pg.16, lines 606-617):

“Periodontal breakdown is found in orthodontic patients, with little available information about the stress distribution under the light orthodontic forces (0.5 N). The clinician needs to know not only the most affected areas by orthodontic stresses but also the amount of stress that could appear during the most stressful orthodontic movements, to be able to individualize the treatment and correlate with bone loss levels. Another practical issue is related to the fact that based on herein results, a single unitary suitable failure criterion for the study of tooth and surrounding periodontium can be seen, strictly correlated with the physical properties and ductile resemblance of dental tissues. The researcher benefits from the herein biomechanical correlations, supplying important data for further studies that are needed for a better understanding of the biomechanical behavior of periodontal breakdown and minimalizing the risks of further tissue loss.”

Reviewer 3 Report

I found the study interesting and timely and the manuscript well organized. However, my primary concern relies on the statement "Periodontal breakdown is a common problem in orthodontic patients, affecting the biomechanical behavior of the tooth and surrounding support tissues, with higher tissular amounts of stress appearing along with bone loss [2, 10-12]", with no mention to periodontal health/pathology before and during orthodontic treatment.

Please, expand on this topic throughout manuscript sections, including periodontal health status and biofilm in relation to orthodontic tooth movement with fixed appliances and clear aligners, which is extremely relevant with regard to periodontal breakdown.

Avoid using bullet points in the Conclusion section.

Author Response

Department of Cariology, Endodontics and Oral Pathology

School of Dental Medicine

University of Medicine and Pharmacy

Ms. Amelia Men

Assistant Editor

Medicina        

Special Issue - Dentistry: A Multidisciplinary Approach  

                                                                                                                                 August 6th, 2023

Dear Ms. Amelia Men,

Thank you very much for your letter dated August 5th, 2023, with the comments of the reviewers. We have now carefully considered the comments of the reviewers and amended the paper accordingly. All changes are highlighted in red throughout the manuscript and included also below.

Reply to Reviewer #3:

We agree and we thank the reviewer for his/her time and comments. Appropriate changes in the manuscript have by now been made. Please see below and in the manuscript.

Concern of the reviewer:

” Comments and Suggestions for Authors

I found the study interesting and timely and the manuscript well organized. However, my primary concern relies on the statement "Periodontal breakdown is a common problem in orthodontic patients, affecting the biomechanical behavior of the tooth and surrounding support tissues, with higher tissular amounts of stress appearing along with bone loss [2, 10-12]", with no mention to periodontal health/pathology before and during orthodontic treatment.

Please, expand on this topic throughout manuscript sections, including periodontal health status and biofilm in relation to orthodontic tooth movement with fixed appliances and clear aligners, which is extremely relevant with regard to periodontal breakdown.

Avoid using bullet points in the Conclusion section..”

Point-by-point response to the reviewer’s comments:

  1. Concern of the reviewer:

“I found the study interesting and timely and the manuscript well organized. However, my primary concern relies on the statement "Periodontal breakdown is a common problem in orthodontic patients, affecting the biomechanical behavior of the tooth and surrounding support tissues, with higher tissular amounts of stress appearing along with bone loss [2, 10-12]", with no mention to periodontal health/pathology before and during orthodontic treatment.

Please, expand on this topic throughout manuscript sections, including periodontal health status and biofilm in relation to orthodontic tooth movement with fixed appliances and clear aligners, which is extremely relevant with regard to periodontal breakdown.”

Our response:

  • We thank the reviewer for his/her concern and comments. We do hope that our changes are according to the reviewer‘s remarks.

The aims of this study were to assess the biomechanical behavior of bone during a gradual horizontal periodontal breakdown from 0 to 8 of loss, as well as finding the proper failure criteria. The patients included in the study (nine) had various levels of bone loss primary in the cervical third and non-inflamed periodontium. However, the initial models with bone loss were reconstructed to no bone loss state, and then the gradual horizontal bone loss was simulated. Thus, the “periodontal health/pathology before and during orthodontic treatment” was not considered for this study and does not change the aims or the results and conclusions. The topic of “periodontal health status and biofilm in relation to orthodontic tooth movement with fixed appliances and clear aligners” mentioned by the reviewer was not among the aims of herein study and has no influence over the results and conclusions of the study. In support of above, please find some relevant parts of the manuscript:

Revised text: pg.4 lines 176-181

“The objectives of herein FEA analysis are: a. to assess the biomechanical behavior of mandibular bone subjected to light orthodontic forces during a horizontal periodontal breakdown; b. to assess the suitability for the study of bone of five of the most used failure criteria employed in dental tissues research; c. to correlate the results with other FEA related reports of dental tissues for identifying a single unitary suitable failure criteria for the study of tooth and surrounding periodontium.”

Revised text: pg. 4 lines 192-198

“The selected convenience sample size of nine was acceptable for the accuracy of the results since most of the earlier FEA studies employed a sample size of one (one patient, one 3D model and few simulations) [1, 2, 5, 8, 10-22, 26-40]. The research project inclusion criteria were the presence of non-inflamed periodontium and various levels of bone loss, intact arch and second premolar tooth structure, lack of endodontic treatment and malposition in region of interest, indication of orthodontic treatment, regular follow-up.””

Revised text: pg.5 lines 207-221

“Thus, enamel, dentine, dental pulp, neurovascular bundle, periodontal ligament, cortical and trabecular bone were 3D reconstructed (Figure 1). The reconstruction software was Amira 5.4.0 (Visage Imaging Inc., Andover, MA, USA). The base of the bracket assumed to be of stainless steel was reconstructed on the vestibular side of the enamel crown. Since the separation of dentine and cementum was impossible, and due to similar physical properties, the entire dentine-cementum structure was reconstructed as dentine (Table 1). The PDL had a variable thickness of 0.15-0.225 mm and included the NVB of the dental pulp in the apical third. The 3D models guarded only the second lower premolar, while the other tooth structures were replaced by cortical and trabecular bone. The missing bone and PDL (which were found in the cervical third) were reconstructed as close as possible to the anatomical reality. Thus, nine models with intact periodontium (one from each patient) were obtained. In each of these models a gradual horizontal breakdown process (0-8 mm of loss) was simulated, by reducing both bone and PDL by 1 mm, obtaining a total of eighty-one models with various levels of bone loss.”

Revised text: pg.2 lines 63-72

“The orthodontic stresses from the tooth are transmitted through the PDL to the bone, while the display areas are also influenced by the tissular anatomy and integrity [8]. Periodontal breakdown is found in orthodontic patients, affecting the biomechanical behavior of the tooth and surrounding support tissues, with higher tissular amounts of stress appearing along with bone loss [2, 10-12]. In bone structure Burr et al. [7] reported microcracks and microdamage near the resorption and remodeling areas, and a decrease of fracture risks in their presence (due to internal micro-architecture changes), which influence the structural stress display. It must be emphasized that the biomechanical behavior in both intact and reduced periodontium is multifactorial, depending on cortical and trabecular structural continuum, material, and structural properties [6]..”

Revised text: pg.4 lines 182-189

“Herein numerical analysis is part of a larger stepwise research project (clinical protocol 158/02.04.2018) continuing the investigation of biomechanical behavior of tooth and surrounding periodontal structure during orthodontic movements and various levels of periodontal breakdown.

The earlier analyses of this project with focus on the dental pulp, neuro-vascular bundle (NVB), periodontal ligament (PDL), dentine and enamel were conducted using the same models, boundary conditions and physical properties as here [2, 10-12].”

Revised text: pg.1 lines 17-37

Abstract: This numerical analysis investigated the biomechanical behavior of the mandibular bone as structure subjected to 0.5 N of orthodontic force during periodontal breakdown. Additionally, assessed the suitability of the five most used failure criteria [Von Mises (VM), Tresca (T), Maximum Principal (S1), Minimum Principal (S3), and Hydrostatic Pressure (HP)] for the study of bone and identify a single criterion for the study of tooth and surrounding periodontium (by performing correlations with other FEA studies). The finite elements analysis (FEA) employed 405 simulations over eighty-one mandibular models with variable levels of bone loss (0-8 mm) and five orthodontic movements (intrusion, extrusion, tipping, rotation, and translation). For the numerical analysis of bone, the ductile failure criteria are suitable (both T and VM adequate for the study of bone), with Tresca more suited. S1, S3 and HP criteria due to their distinctive design dedicated to brittle materials and liquids/gas only occasionally correctly described the bone stress distribution. Only T and VM displayed a coherent and correlated gradual stress increase pattern for all five movements and levels of the periodontal breakdown. The quantitative values provided by T and VM were the highest (for each movement and level of bone loss) among all five criteria. MHP (maximum physiological hydrostatic pressure) was exceeded in all simulations since the mandibular bone is anatomically less vascularized, and the ischemic risks are reduced. Only T and VM displayed a corelated (both qualitative and quantitative) stress increase for all five movements. Both T and VM displayed rotation and translation, closely followed by tipping as stressful movements, while intrusion and extrusion were less stressful for the mandibular bone. Based on correlations with earlier numerical studies on the same models and boundary conditions, T seems better suited as single unitary failure criterion for the study of tooth and surrounding periodontium.”

Revised text: pg.16 lines 606-617

“Periodontal breakdown is found in orthodontic patients, with little available information about the stress distribution under the light orthodontic forces (0.5 N). The clinician needs to know not only the most affected areas by orthodontic stresses but also the amount of stress that could appear during the most stressful orthodontic movements, to be able to individualize the treatment and correlate with bone loss levels. Another practical issue is related to the fact that based on herein results, a single unitary suitable failure criterion for the study of tooth and surrounding periodontium can be seen, strictly correlated with the physical properties and ductile resemblance of dental tissues. The researcher benefits from the herein biomechanical correlations, supplying important data for further studies that are needed for a better understanding of the biomechanical behaviour of periodontal breakdown and minimalizing the risks of further tissue loss.”

  1. Concern of the reviewer:

“Avoid using bullet points in the Conclusion section...”

Our response:

  • We thank the reviewer for his/her concern and comments. We do hope that our changes are according to the reviewer‘s remarks.

Revised text: pg.16 lines 619-636

“For the numerical analysis of bone, the ductile failure criteria are suitable (both T and VM adequate for the study of bone), with Tresca more adequate than VM.

The S1, S3 and HP failure criteria due to their distinctive design dedicated to brittle materials and liquids/gas only occasionally correctly described the distribution of stresses in the bone.

For all five orthodontic movements and bone levels only T and VM displayed a coherent and correlated gradual stress increase pattern along with the periodontal breakdown.

The quantitative values provided by T and VM were the highest among all five criteria (for each movement and level of bone loss).

MHP (maximum physiological hydrostatic pressure) was exceeded in all simulations, since the mandibular bone is anatomically less vascularized, and the ischemic risks are reduced.

T and VM displayed rotation and translation, closely followed by tipping as stressful movements, while intrusion and extrusion were less stressful for the mandibular bone.

Based on correlations with other earlier numerical studies on the same models and boundary conditions, T could be seen as a single unitary suitable failure criterion for the study of tooth and surrounding periodontium.” 

Round 2

Reviewer 1 Report

The authors have made the requested corrections.